# Creation of a Pilot School Health Research Network in an English Education Infrastructure to Improve Adolescent Health and Well-Being: A Study Protocol

**DOI:** 10.3390/ijerph192013711

**Published:** 2022-10-21

**Authors:** Catherine A. Sharp, Emily Widnall, Patricia N. Albers, Kate Willis, Colin Capner, Judi Kidger, Frank de Vocht, Eileen Kaner, Esther M. F. van Sluijs, Hannah Fairbrother, Russell Jago, Rona Campbell

**Affiliations:** 1Population Health Sciences, Bristol Medical School, University of Bristol, Bristol BS8 2PS, UK; 2Faculty of Medical Sciences, Newcastle University, Newcastle upon Tyne NE1 7RU, UK; 3MRC Epidemiology Unit, University of Cambridge, Cambridge CB2 0SL, UK; 4Health Sciences School, University of Sheffield, Sheffield S10 2TN, UK; 5Centre for Exercise Nutrition & Health Sciences, School for Policy Studies, University of Bristol, Bristol BS8 1TZ, UK

**Keywords:** adolescence, mental health, well-being, school health research network, systems intervention

## Abstract

Schools play a significant role in promoting health and well-being and the reciprocal links between health and educational attainment are well-evidenced. Despite recognition of the beneficial impact of school-based health improvement programmes, significant barriers to improving health and well-being within schools remain. This study pilots a School Health Research Network in the South West of England (SW-SHRN), a systems-based health intervention bringing together schools, academic health researchers and public health and/or education teams in local authorities to share knowledge and expertise to improve the health and well-being of young people. A maximum of 20 secondary schools will be recruited to the pilot SW-SHRN. All students in Years 8 (age 12–13) and 10 (age 14–15) will be invited to complete a health and well-being questionnaire, generating a cohort of approximately 5000 adolescents. School environment questionnaires will also be completed with each school to build a regional picture of existing school health policies and programmes. Each school will be provided with a report summarising data for their students benchmarked against data for all schools in the network. Quantitative analysis will model associations between health risk behaviours and mental health outcomes and a qualitative process evaluation will explore the feasibility and sustainability of the network. This study will create adolescent health data to help provide schools and local authorities with timely and robust information on the health and well-being of their students and help them to identify areas in which public health interventions may be required. SW-SHRN will also help public health professionals focus their resources in the areas most at need.

## 1. Introduction

There is an inextricable link between health and education with evidence suggesting that attending to health in schools improves educational attainment [1]. Children who are healthy learn more readily and educational attainment is associated with living longer, healthier, and happier lives [2]. Additionally, children’s emotional health and well-being is associated with their cognitive development and capacity to learn, as well as with their physical health, social well-being, and mental health in adolescence and adulthood [3]. While children and young people in the United Kingdom (UK) generally report relatively high levels of well-being [4], there is evidence of secular increases in the prevalence of mental health and emotional disorders [5] with added concerns for young people’s mental health due to the COVID-19 pandemic [6,7,8]. Adolescence is a critical development period defined by substantial physical, behavioural, and social changes and therefore offers a key opportunity for early intervention to promote health and well-being across the life course [9].

Schools can play an important role in promoting emotional health and well-being. Early intervention and support for those who develop mental, emotional, and behavioural disorders generally improves the chances of a better outcome [10]. Schools are a pivotal setting for young people’s health improvement with increasing evidence of the relationship between the school environment and student health [11]. More than two decades ago, the World Health Organization (WHO) advocated a whole-school approach to health improvement through the Health Promoting Schools Framework [12]. The three core elements of this framework comprise: (i) health education being addressed within the school curriculum; (ii) promotion of health and well-being through changes to the schools’ social and/or physical environment; and (iii) schools engaging with families and communities to reinforce health messages beyond the school environment. A Cochrane systematic review has shown the framework to be effective in encouraging healthy behaviours such as physical activity and healthy eating, and in preventing health damaging behaviours such as tobacco use and bullying [13]. In 2019, the WHO renewed calls internationally to make every school a health promoting setting [14]. In the UK, Public Health England produced guidance to help and encourage schools to recognise the link between health and educational attainment and to promote children and young people’s emotional health and well-being in schools [3].

While there is a welcome reduction in the prevalence of some health risk behaviours in young people in the UK, notably in tobacco, alcohol, and drug use, increased rates of anxiety and depression, and low levels of physical activity, remain a cause for concern [15]. Furthermore, recent research has shown a strong association between engagement in multiple risk behaviour and substantial adverse health and social outcomes in early adulthood [16], and poorer educational attainment at secondary school at age 15 to 16 years [17]. In addition, adolescent multiple risk behaviour has been shown in two UK cohort studies to be negatively associated with socio-economic status in adulthood as indicated by University degree attainment [17,18,19]. Significant associations have also been found between psychological distress and multiple risk behaviour in adolescence [20] indicating a continued need to work to reduce health risk behaviours and improve mental health [16].

A variety of health and well-being surveys are conducted in UK schools. These include Local Authority (LA) commissioned surveys through private companies as well as schools designing and delivering internal surveys. Additionally, cross-national longitudinal surveys exist such as the Health Behaviour in School-aged Children (HBSC) which collects health and well-being data every four years on 11-, 13-, and 15-year old students. In England the HBSC provides national-level data for approximately 4000 students derived from a sample of 33 secondary schools [21]. These surveys are helpful in indicating the overall prevalence of health and social risk behaviours and other key health indicators for the school population, but there is a need for school-specific data to provide school-specific action and interventions.

Despite growing recognition of school-based health improvement, there remain barriers to improving health and well-being within school settings in England. These include the challenge of balancing the promotion of academic attainment with efforts to ensure student and staff health and well-being [1], and a lack of integration between school health research and the core business of schools, which is to educate their students and to reduce inequities in attainment. One established method for overcoming these barriers and imbalances has been the creation of school health research networks (SHRNs). Established SHRNs exist within the UK such as the School Health Research Network in Wales; https://www.shrn.org.uk/ (accessed on 6 June 2022) and the Schools Health and Wellbeing Improvement Research Network (SHINE) in Scotland; https://shine.sphsu.gla.ac.uk/ (accessed on 6 June 2022). School health research networks use a whole system approach to facilitate health improvement in schools which bring together stakeholders and communities to develop a shared understanding of how best to improve child health and well-being. System approaches encourage continuous partnership working and knowledge exchange to create system-wide changes and a data-rich resource that multiple stakeholders in young people’s health and education can use to inform their work [22]. A regional school health research network has yet to be developed in England.

The education system in England is unique and provides distinct challenges to building a school health research network. The types of school are far more diverse than those in the other UK nations. This is the result of policy to reduce the number of local authority-maintained schools in favour of state funded but independent academy trust schools, and free schools, which report directly to the Department for Education. Currently, only 698 of 6142 (11.4%) secondary schools in England are local authority maintained. As part of their greater independence, academy schools do not have to follow the national curriculum and the structures and scope for influencing what schools implement in relation to health has been diminished. In 2019, there were 1170 multi-academy trusts (MATs), but even within schools belonging to MATs, research suggests the individual schools retain responsibility on how they approach health improvement within schools; with little direction and policy being provided by senior MAT management [23]. Wide variability was also found in the approach to promoting student’s health across academy trusts and academy schools with some even indicating that they thought they held no responsibility for health improvement because their core business was education and not health [23]. There have also been a number of changes in government policy around school health in England, namely the ending of school health policies pursued under the Labour government as well as the Department for Education introducing compulsory relationships and sex education (RSE) and health education for secondary students in 2020 [24]. Responding to these changes, local and national public health and education teams have expressed a desire to create more effective working practices with schools in England to improve adolescent health outcomes [23].

The aim of this study is to pilot a school health research network in the South West of England which is intended to form the basis for a larger regional network. In the long-term, this systems-level intervention aims to produce a cohort of school-aged children from Year 8 and Year 10 that provides robust health and well-being data for schools and local, regional and national stakeholders. A process evaluation will also allow schools and key stakeholders to provide meaningful insight into the facilitators and barriers to setting up a regional school health research network in England as well as providing important suggestions on the networks impact and sustainability. Young people’s responses will be individually tracked overtime to allow meaningful within-person longitudinal comparisons. This would, over time, allow the network the opportunity to help monitor the effect of any school-based interventions, policies and practices that are implemented in response to findings of the school-based surveys.

This paper outlines the protocol for creating, piloting, and evaluating the South West—School Health Research Network.

## 2. Materials and Methods

### 2.1. Aims

The overall aim of this project is to pilot a SHRN amongst secondary schools in the South West of England to improve the health and well-being of adolescents. This will be achieved through the following seven objectives:(1)Work with stakeholders to co-produce the key policies and processes for the creation of a SHRN in the South West of England.(2)Recruit a maximum of 20 secondary schools to participate in the first phase of the SW-SHRN and collect information on mental health and health risk behaviour programmes and policies through a school environment survey.(3)Recruit up to 2500 Year 8 and 2500 Year 10 students within the recruited schools, and collect self-reported data from the students on mental health and health risk-taking behaviours.(4)Examine the associations between students’ health risk-taking behaviours and mental health.(5)Identify the extent to which any associations are modified by school health promotion policies, school characteristics (Ofsted rating, Pupil Premium and progress scores and school location), and student characteristics such as age, gender, and social economic status.(6)Identify stakeholder views of a regional network model, identify key issues that impact (positively or negatively) the implementation of a regional SHRN in England and identify key barriers and facilitators to scalability and sustainability.(7)Further develop and refine our working logic model and programme theory for the SW-SHRN.

### 2.2. Design

SW-SHRN is a pilot study of a SHRN. It is intended that this pilot study will provide a new method of partnership working at a regional- and LA-level between public health researchers, public health practitioners, school educators, students, and parents; all of whom are key stakeholders in the health and well-being of adolescents. This systems-level, school-based intervention will utilise mutual influence, parallel processes, and feedback loops, to affect the delivery of public health in participating schools and associated policies aimed at improving adolescent health and well-being. These processes will allow the evaluation to attend to the contexts in which the SW-SHRN is implemented and received, including the effects on the broader functioning of the social ‘systems’ operating within different contexts [25]. Ongoing collaboration between researchers, the public and public health policy makers and practitioners will also maximise the relevance, acceptability, and scale-ability of the network [26]. Whilst we have developed a working logic model (see Appendix A), this pilot study aims to build and refine a programme theory [27] to better understand how the systems-based intervention works, under what conditions, and how key components of the intervention interact. A defined programme theory will then support refinement of our logic model.

Schools in the SW-SHRN will complete a health and well-being survey with Years 8 and 10 students. We will also use a school environment survey to establish the health and well-being policies and programmes that are in place within the schools. SW-SHRN will generate a robust cohort of adolescent health data and school health policy information, to both facilitate knowledge sharing between key stakeholders and allow evidence-based identification of key focus areas for schools to inform intervention development. Figure 1 illustrates SW-SHRN’s operational model.

### 2.3. Study Population

The study population will comprise of Year 8 (aged 11–12) and Year 10 (aged 13–14) students in secondary schools across the South West of England. We will also work with the relevant school staff (e.g., senior leadership and mental health and well-being leads), and a range of key stakeholders (e.g., government departments and charities) involved in education or health promotion in schools.

### 2.4. Inclusion Criteria

#### 2.4.1. School Surveys

Secondary schools must be within one of the fifteen LAs (see Appendix A) within the South West of England.Students completing the health and well-being survey must be in Years 8 or 10.Staff completing the school environment survey must have sufficient knowledge of school health improvement policies and interventions.

#### 2.4.2. Qualitative Interviews

The key school contact involved in the organisation and delivery of the SW-SHRN survey.Stakeholders involved in education and/or health promotion in schools.

### 2.5. Exclusion Criteria

Special schools and pupil referral units will not be approached as study schools in this pilot study. Such schools lie outside of the majority school provision in England (Department for Education, 2019), and will be better suited to further study once a general regional SHRN model has been created and tested.

### 2.6. Sample Size

#### 2.6.1. Quantitative Component

A maximum of 20 schools will be recruited from across the South West of England with the aim of engaging a minimum of six of the 15 LA areas to increase representativeness from across the LAs. A greater representation of different LA areas will reduce clustering effects emerging from LA areas. While LAs no longer have responsibility for schools their past decisions will have influenced the current pattern of school provision and management in their area and LAs still provide support for health promotion in schools. Therefore, schools in one LA area are likely to be more similar to each other than to schools in other LAs. The student survey sample size will consist of approximately 2500 Year 8 students and 2500 Year 10 students providing a total dataset of up to 5000 students.

#### 2.6.2. Qualitative Component

Approximately 30 qualitative interviews will be conducted. One key contact from each participating school will be invited to participate in a qualitative interview (up to 20 school staff interviews). Additionally, to provide a regional and national perspective on the SW-SHRN, approximately 10 key stakeholders (education and public health leaders from LAs, the Office for Health Improvement and Disparities, Ofsted and Departments of Health & Social Care and Education) will be invited to participate in an interview. A list of key stakeholders relevant to public health education in school will be developed by the research team in collaboration with policy and practice partners.

### 2.7. Measures

#### 2.7.1. Student Health and Well-Being Survey

The student survey is designed to capture a wide breadth of public health topics relevant to the adolescent developmental period. Due to time pressures in school, the survey has been designed to be completed in a maximum of 45 min to fit into a single lesson. The survey does not contain any items that would trigger a safeguarding response; student answers are therefore confidential.

A composite questionnaire has been created that assesses mental health and well-being, physical activity and eating behaviour, sexual health, risky behaviours including smoking and alcohol use, body image, sleep, peer support, cyberbullying, social media use and the school environment. Items were drawn from the HBSC Survey, the Welsh SHRN and physical activity items from the Youth Activity Profile. In addition, the survey contains the following scales; Short Warwick-Edinburgh Mental Well-being Scale (SWEMWEBS) to assess mental well-being [28], Short Moods and Feelings Questionnaire (SMFQ) to assess moods and feelings [29], the Generalized Anxiety Disorder Scale (GAD-7) to assess anxiety [30,31], and the Child Health Utility-9 Dimensions (CHU9D) to assess quality of life [32].

Years 8 and 10 students complete the same survey, but only Year 10 students are asked questions relating to sexual relationships. Public health colleagues and young people were consulted on the content, and young people subsequently piloted the questionnaire (see Public Involvement).

#### 2.7.2. School Environment Survey

The school environment survey assesses what current health policies and interventions schools already have in place as well as gathering information on teaching responsibilities and curriculum time allocated to health and well-being. The health policy areas are in line with the health areas covered within the student survey (mental health, physical activity, healthy eating, smoking, drug and alcohol use, social well-being, sex and relationships). The survey primarily consists of closed-response, multiple choice questions but also free-text responses to allow schools to list and describe details of existing health interventions in place. Typically, a member of the senior leadership team will complete the school environment survey, or alternatively, a mental health lead with sufficient knowledge of the school’s health and well-being policies.

#### 2.7.3. Stakeholder Interviews

Topic guides will cover how key stakeholders can support the network, what barriers and facilitators they foresee to developing such a network, what outputs they would like to see from the network and their views on how to create a sustainable and scalable network. Additionally, key school contacts within participating schools will be asked about their experiences participating in the network in terms of logistics of administering the student surveys, their views on receiving tailored school reports of student data from the surveys, how they would use the data provided by the network to create improvements within their school, and what would encourage them to continue being part of the network on a longer-term basis.

### 2.8. Procedure

The study procedure consists of co-production of processes and data collection tools, school recruitment, data collection and report production, qualitative interviews with key stakeholders, and data analysis and dissemination activities. Figure 2 outlines the procedural stages of the creation, piloting, and evaluation of the SW-SHRN.

#### 2.8.1. School Recruitment

In line with a systems-level approach to engage all relevant stakeholders in the process, LAs will first be approached and recruited to engage with the SW-SHRN. Contact will be made with those concerned with the delivery of public health policy and/or education in schools. Where possible, the contact will help facilitate communication between the SW-SHRN and schools. An information brochure summarising the offer of SW-SHRN will be shared with prospective LAs and schools. A second recruitment approach will be to contact schools and academy trusts directly across the 15 LA areas. The research team will endeavour to make contact directly with school’s senior leadership teams where possible.

Following each schools’ agreement to participate, the school will be asked to electronically co-sign the following agreement documentation:(1)Research agreement—this agreement outlines the commitment from the research team and expectations of participating schools.(2)Data sharing agreement—in compliance with requirements of General Data Protection Regulation (GDPR), this agreement details the data processing of personal data (students’ name, date of birth and postcode) as well as how the data will be used to inform and improve public health provision. This agreement also details the purpose of collecting a small amount of personal data to allow linkage of individual responses overtime to enable the research team to monitor longitudinal changes as interventions are introduced to evaluate their impact on student health and well-being.(3)User agreement—this agreement outlines use and sharing of any data reports produced by SW-SHRN. LAs will also be required to co-sign a user agreement.

#### 2.8.2. School Reimbursement

Each school will receive a gratuity payment of £200 for participating in the network to provide recompense for staff time facilitating the data collection. Schools who conduct the survey during school closures (due to the COVID-19 pandemic) will be offered a higher gratuity payment of £300 to reflect the additional level of administrative support required from schools.

#### 2.8.3. Public Involvement

A project specific Young Person’s Advisory Group (YPAG) will be established from the NIHR Applied Research Centre West’s existing YPAG. The group will meet online biannually and consist of up to eight young people from Years 7 to 10, with a mixture of boys and girls. The young people will be asked to provide input into the survey content, wording of questions, feedback reports, data collection processes in the schools, and the digestibility of outputs (e.g., infographics) provided to the schools. Their views will feed into the Stakeholder Group Meetings. Members of this YPAG will receive a £20 gift voucher for each hour that they participate.

Teachers and headteachers will also be invited to contribute to the Stakeholder Advisory Group meetings or to support recruitment efforts via LAs. To ensure the network is centred to the needs of the schools and LAs, the Stakeholder Advisory Group will include colleagues from the LA, Department for Education, Office for Health Improvement and Disparities, and a public representative. Public stakeholders will advise on the design and implementation of the network as well as on best practices for the dissemination of findings.

#### 2.8.4. Survey Implementation

##### Student Survey

The survey will primarily be delivered online via an electronic data capture tool; REDCap (Research Electronic Data Capture), a secure web-based software platform hosted by the University of Bristol [33,34]. Using R [35] (a statistical software package), a function will be written to generate and assign each student a unique identification code. Students will be allocated a unique access code, created within the REDCap system, that is linked to their unique identification code. This ensures no personal data is included in the student survey data sets and allows individual student respondents’ responses to be linked overtime, if they participate in future surveys and provide consent to do so.

Due to COVID-19 restrictions the following flexible data collection methods were offered to early adopter schools which are listed in order of preference from an expected completeness of data perspective:(1)Classroom-based with a researcher and teacher present.(2)Classroom-based with only a teacher present (no researcher).(3)Remote live teaching session with a researcher overview present via live video link.(4)Remote non-live scheduled teaching session with a researcher overview video.(5)Remote independent task for students to complete survey outside of timetable.

For classroom-based sessions, schools will have the option to use SW-SHRN provided tablets or their own electronic devices (e.g., computers/tablets). Paper copies of the survey can be provided if requested by the school. Students will be directed to support resources hosted on the SW-SHRN website should they require any further information or support on the topics covered in the survey.

The survey does not include any specific questions that might create a safeguarding concern. Confidentiality will be maintained within the study, unless an individual (student or adult) discloses information to the researcher which suggests they or others are at-risk of serious harm. “At-risk of serious harm” will be considered as any information which relates to any sort of abuse (physical, sexual, psychological, neglect, financial, discriminatory) or of serious self-harm. If this occurs, individual school safeguarding procedures will be followed. In addition, at the end of the project, an anonymised summary of any disclosures will be reported to the Project Management Group and the approving ethics committee.

##### Student Survey Consent Process

The first stage of the consent process is parental opt-out consent. Schools will be required to communicate with parents about the study through a minimum of two approaches: (i) distribution of a parent-specific information letter and (ii) an approach in line with the school’s communication strategy (e.g., school newsletter, e-mail). A withdrawal form will accompany the information letters. Parents/students will be able to withdraw the student’s participation up to two working days prior to the first day of data collection, or up to five working days after data collection is completed, at which point their data will be permanently removed. This short timeframe after data collection is required to enable the rapid analysis of data and compilation and delivery of the school report.

On the day of data collection, students will be asked to provide informed consent electronically prior to completing the survey as well as assent. In addition, they will be asked to provide consent for their data to be connected longitudinally, and for their data to be linked to routinely collected data (e.g., National Student Database). Such linkage will not be undertaken as part of this pilot study, but if it is successful, we would hope to undertake such data linkage as part of the network’s development.

Opt-out consent procedures will also be administered to staff participating in the school environment survey. The staff member will receive a full information sheet prior to participation and electronically provide consent prior to beginning the survey.

Written informed consent will be obtained prior to all qualitative interviews commencing. Where written consent cannot be obtained, verbal consent via audio-recording will be obtained and stored separately from the interview.

##### School Environment Survey

The survey questions will be sent as a PDF file to key school contacts in advance to allow schools to gather data for some specific questions (e.g., pupil premium spending). The primary method of data collection will be an online structured interview with a member of the research team who will go through each question with the member of school staff. This will allow clarification and discussion around the responses given to ensure accurate responses are recorded. Staff responses will be directly captured by a researcher into a data capture form hosted on the REDCap survey system.

#### 2.8.5. Knowledge Transfer

##### Tailored School Reports with Benchmarking

Each school will receive a summary report and infographic of their student’s data approximately four weeks after data collection. Schools will be encouraged to share the findings with students and parents. The output will be accompanied with the offer of an audio or video call with a researcher to discuss the findings to maximise on knowledge transfer and accurate interpretation, as data literacy of the contact person will be unknown. Once sufficient student survey data across the entire study have been collected, each school will be provided with an updated report containing benchmarked data from other schools in the network. These reports will allow schools to identify areas for health improvement as well as identifying areas of strength that may need less focus. Benchmarking will therefore enable schools to prioritise efforts to key target areas.

##### School Environment Report

After all schools complete the school environment survey, a summary report and infographic of aggregated data will be provided to each school. This report will provide an overview of current health improvement programmes run within the network schools as well as what health and well-being policies schools currently have in place. Providing an overview of existing health interventions and policies will facilitate shared learning across participating schools.

##### LA Summary Reports with Benchmarking

Each LA will receive a summary report on the data collected from schools within their area, with benchmarking provided from all participating schools. LA reports will also provide a summary of the school environment data collected from schools in their area.

## 3. Data Analysis

### 3.1. Quantitative Analysis

Descriptive statistics (including means, standard deviations, frequencies, and percentages) will be calculated for the student and school environment surveys. These descriptive analyses will be used to characterise the distribution of policies by school features such as size, Ofsted rating, level of deprivation, and school location (e.g., urban vs. rural). We will explore differences in survey responses between LAs and schools taking into account potential inequalities such as disabilities or the proportion of students eligible for free school meals. We will also explore which data collection method had the highest response rates.

These cross-sectional data will be used to explore associations between risks, such risk-taking behaviours or school environment characteristics, and health outcomes. For instance, poor mental health will be explored using mixed effects logistic regression or mixed effects multinomial logistic regression. These models will include possible confounders and other interactions between different exposures. Since student responses from a school or LA may be more similar to each other than responses from other schools or LAs, the data will be treated as clustered at school and LA levels. This clustering can be accounted for in the mixed effects models, whereby the model allows responses to differ by cluster, known as random effects, and therefore not skew the relationship of interest (fixed effect [36,37]).

Other statistical tests or forms of analysis may be used as appropriate to the data with details specified in the final reports. Analyses will be run for all participants and may then be stratified by sub-group, such as age or gender, and modelled with interaction terms to test for associations within sub-groups.

### 3.2. Qualitative Analysis

Qualitative interviews will be audio-recorded and transcribed verbatim. NVivo 12 will be used to support data management and analysis. The transcripts will be analysed using the Framework Method in which thematic analysis is conducted and themes are developed, both inductively from the accounts of participants and deductively from existing literature [38]. Brief summaries, mind maps, and representative quotes for each category will be abstracted and compared across participant groups. Data will then be synthesised to provide key guidance on the running of the network, sustaining the network, and creating further networks in other areas of England.

## 4. Discussion

This paper presents the protocol for the creation and evaluation of a SHRN in the South West of England. Data collected through the SW-SHRN will help provide schools and LAs with timely and robust data on the health of their students and help them to identify areas in which public health intervention may be required. This, in return, will help public health partners focus their resources on the areas of most need. Additionally, the SW-SHRN seeks to create a new regional method of partnership working in England bringing together schools, academy trusts, academic health researchers, and public health and education teams in LAs. It is intended that the study will provide key data on the potential for creating a SHRN in England and that the data collected in this pilot study could form the foundation for future data collections which would facilitate the creation of a new cohort design. This would involve linked biennial surveys to track changes in adolescent health and well-being outcomes over time as well as the ability to track the impact of any newly implemented health and well-being interventions within participating schools.

## 5. Conclusions

SW-SHRN will combine accurate identification of health challenges that schools face, sharing knowledge of evidence-based policies and interventions to improve student health and using repeated surveys to monitor and feedback the impact of any changes made by the schools. This combined approach has the potential to effect significant improvements in both health and educational outcomes.

Although similar SHRNs exist in both Wales and Scotland, SW-SHRN will be the first regional school health research network in England as well as the first to offer regional benchmarking data to schools and LAs.

## Figures and Tables

**Figure 1 ijerph-19-13711-f001:**
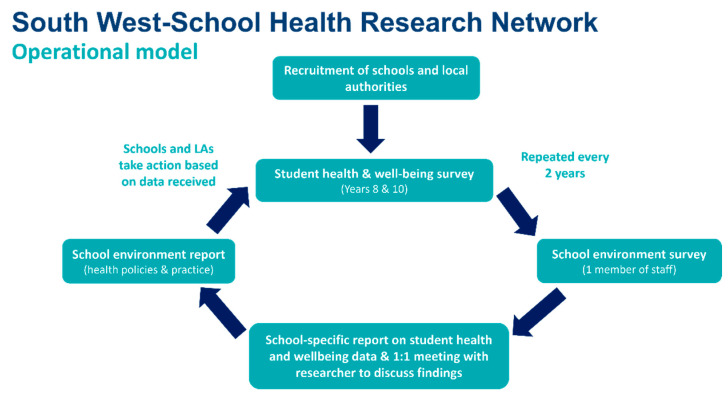
SW-SHRN Operational Model.

**Figure 2 ijerph-19-13711-f002:**
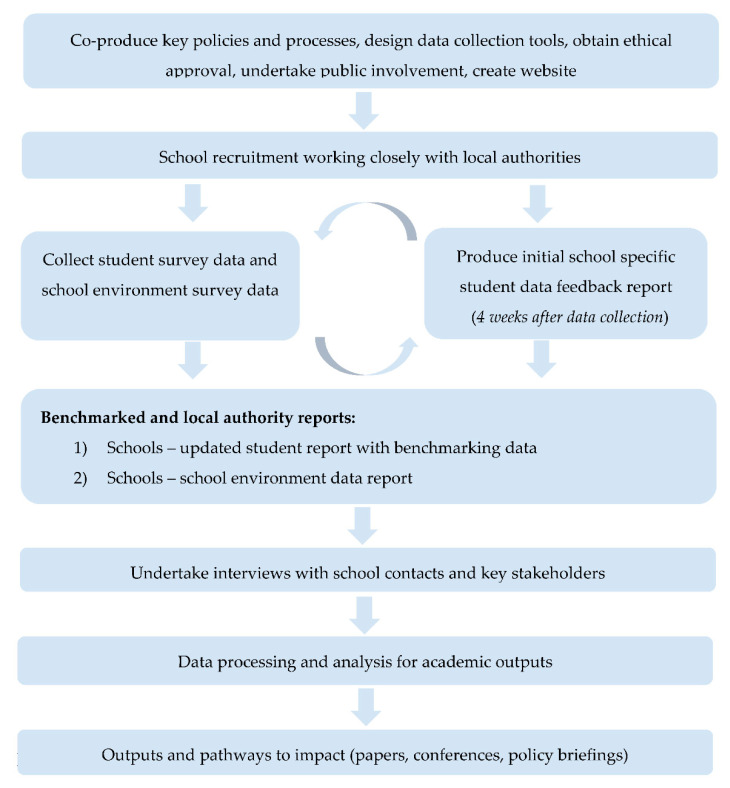
SW-SHRN Procedural Overview.

## Data Availability

Not applicable.

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
