# Peer review of "Creation of a Pilot School Health Research Network in an English Education Infrastructure to Improve Adolescent Health and Well-Being: A Study Protocol"

_ijerph, 2022, doi:10.3390/ijerph192013711_

Round 1

Reviewer 1 Report

This is more or less a research proposal rather than a research article. Nothing has been done in this article. It looks some undergraduate work and they have just submitted it in the journal. From my point of view it is not suitable to publish in such a good impact factor journal.

Author Response

We respectfully disagree with the reviewer. This is a study protocol which specifies the research plan for the pilot study and therefore the manuscript contains no findings. This approach is considered best practice for large studies and is consistent with open science principles. This is a large pilot study led by two Professors of Public Health. The study has been peer reviewed by the National Institute of Health Research’s, School for Public Health Research (NIHR SPHR).

Reviewer 2 Report

Thank you for the opportunity to review this study protocol. I would like to congratulate the authors on a well-detailed manuscript. The implementation of the described project is certainly very desirable and may bring many benefits for the health of adolescents, their well-being and the school environment.

I have only a few minor comments on the manuscript:

- Line 149: You state that the main aim will be achieved through 6 objectives, but below is a listing of 7 sub-objectives.

- Line 167: You state that you want to "develop a logic model". However, this already appears to have been developed. Perhaps the objective could be modified to state that the model is to be verified and refined.

- Line 205: You state that "Staff completing the school environment survey must have sufficient knowledge of school health improvement policies and interventions". Is any procedure being considered as to how this will be verified/ensured? If this staff member is the only source of information about the school, every effort should be made to obtain valid data and minimize bias that could seriously impact the entire study.

- Line 258: I see a certain limitation in the validity of the "School environment survey". Could you please add what measures will be put in place to make the responses of school staff as objective as possible? I would recommend to state already in section 2.7.2 that it will be a structured interview (as later mentioned in section 2.8.4). Here the text gives the impression that staff will only provide written answers.

- Line 488: I would recommend leaving only the first sentence in the Institutional Review Board Statement section and moving all other information about student recruitment to the body of the paper. This information is substantial and I do not think it is common practice to include it in this section.

- Line 511: The Informed Consent Statement section states "Not applicable". However, the preceding paragraph contains this statement. I would recommend it be moved here.

I also noticed a few more formal details:

- Line 6: A comma is missing between the authors Fairbrother and Jago.

- Line 6: Russell Jago has affiliations 1 and 5 listed. However, affiliation 5 is not defined below.

- Line 11: The number 4 is used duplicately for two different affiliations.

- Line 532: Reference number 6 duplicates "Association LG" as the author.

Author Response

Thank you for the opportunity to review this study protocol. I would like to congratulate the authors on a well-detailed manuscript. The implementation of the described project is certainly very desirable and may bring many benefits for the health of adolescents, their well-being and the school environment.

The authors thank the reviewer for their positive feedback on the manuscript.

I have only a few minor comments on the manuscript:

- Line 149: You state that the main aim will be achieved through 6 objectives, but below is a listing of 7 sub-objectives.

Corrected

- Line 167: You state that you want to "develop a logic model". However, this already appears to have been developed. Perhaps the objective could be modified to state that the model is to be verified and refined.

Corrected to note we wish to refine our working logic model.

- Line 205: You state that "Staff completing the school environment survey must have sufficient knowledge of school health improvement policies and interventions". Is any procedure being considered as to how this will be verified/ensured? If this staff member is the only source of information about the school, every effort should be made to obtain valid data and minimize bias that could seriously impact the entire study.

Members of staff completing the school environment survey either need to have a specified health and well-being role or be a member of the senior leadership team. The lead researcher discusses who is best to complete the school environment survey ahead of the staff member being decided upon and the survey is shared with the school ahead of completion to they are aware of the content.

A description of staff members is detailed from 267-269:

“Typically, a member of the senior leadership team will complete the school environment survey, or alternatively, a mental health lead with sufficient knowledge of the school’s health and well-being policies.”

- Line 258: I see a certain limitation in the validity of the "School environment survey". Could you please add what measures will be put in place to make the responses of school staff as objective as possible? I would recommend to state already in section 2.7.2 that it will be a structured interview (as later mentioned in section 2.8.4). Here the text gives the impression that staff will only provide written answers.

The following clarification has been added to section 2.84 as this section details how each survey will be delivered:

The primary method of data collection will be an online structured interview with a member of the research team who will go through each question with the member of school staff. This will allow clarification and discussion around the responses given to ensure accurate responses are recorded.

- Line 488: I would recommend leaving only the first sentence in the Institutional Review Board Statement section and moving all other information about student recruitment to the body of the paper. This information is substantial and I do not think it is common practice to include it in this section.

Detailed consent information has now been moved into the main manuscript under the student survey implementation section with a subtitle ‘student survey consent process’.

- Line 511: The Informed Consent Statement section states "Not applicable". However, the preceding paragraph contains this statement. I would recommend it be moved here.

This statement has been updated to:

Informed consent was obtained from all subjects involved in the study

I also noticed a few more formal details:

- Line 6: A comma is missing between the authors Fairbrother and Jago.

Now added

- Line 6: Russell Jago has affiliations 1 and 5 listed. However, affiliation 5 is not defined below.

Corrected

- Line 11: The number 4 is used duplicately for two different affiliations.

Corrected

- Line 532: Reference number 6 duplicates "Association LG" as the author.

Corrected

Reviewer 3 Report

In my opinion, this is a topic that, well raised, can be interesting for educational institutions.

The introduction is not clear or specific, it is not clear what the specific aspect to be analyzed is (it is very ambiguous) and the literature proposed must be more current, so that it adjusts to the current reality. You did an inaccurate research study.

I do not know the questionnaires proposed in the study, nor their validity, and I should have included the questionnaire in the appendix in the article, to see the items that make it up and that you used to do the analyses. If I had it in the appendix, other researchers could use it in future studies. The qualitative part is imprecise and unclear, its process and results are not seen.

The conclusion is not proved, but rather stated without the reader understanding the process by which this claim is reached.

I consider that the article should be clearer, both in the literature and in the methodological part.

Author Response

In my opinion, this is a topic that, well raised, can be interesting for educational institutions.

The authors thank the reviewer for their positive comment on the manuscript.

The introduction is not clear or specific, it is not clear what the specific aspect to be analyzed is (it is very ambiguous) and the literature proposed must be more current, so that it adjusts to the current reality. You did an inaccurate research study.

The authors thank the reviewer for their feedback on the introduction. The School Health Research Network is a systems-based intervention aimed to improve adolescent health and well-being. The introduction therefore covers multiple areas when introducing the network and systems-thinking to include the following:

  • The link between health and education
  • The importance of schools in promoting health and well-being
  • Existing initiatives/frameworks for health improvement in schools
  • Current prevalence of health issues in young people
  • Existing similar approaches (school health surveys)
  • Existing school health research networks (Wales and Scotland)
  • Why a distinct network is needed in England due to it’s complex education system

The specific aims of the study are then detailed in section 2.1.

In terms of making the introduction more current, a number of references from older literature are key to this study protocol, particularly the World Health Organisation Framework (1997), Public Health England’s report (2015) and key literature around health promotion in schools (Bonnell et al.2014).

However, in addition to this key literature, we have now referenced the most recent NHS survey of children and young people’s mental health as well as commenting on the increasing concern for children and young people’s mental health as a result of the COVID-19 pandemic.

The additional literature added to the introduction includes:

  • Newlove-Delgado, T., Williams, T., Robertson, K., McManus, S., Sadler, K., Vizard, T., ... & Ford, T. (2021). Mental Health of Children and Young People in England 2021-wave 2 follow up to the 2017 survey.
  • Lee, J. (2020). Mental health effects of school closures during COVID-19. The Lancet Child & Adolescent Health, 4(6), 421.
  • Power, E., Hughes, S., Cotter, D., & Cannon, M. (2020). Youth mental health in the time of COVID-19. Irish Journal of Psychological Medicine, 37(4), 301-305.
  • Newlove-Delgado, T., McManus, S., Sadler, K., Thandi, S., Vizard, T., Cartwright, C., & Ford, T. (2021). Child mental health in England before and during the COVID-19 lockdown. The Lancet Psychiatry, 8(5), 353-354.

I do not know the questionnaires proposed in the study, nor their validity, and I should have included the questionnaire in the appendix in the article, to see the items that make it up and that you used to do the analyses. If I had it in the appendix, other researchers could use it in future studies.

The following details of the survey content are provided in the manuscript. The authors detail and reference all validated questionnaires where possible.

A composite questionnaire has been created that assesses mental health and well-being, physical activity and eating behaviour, sexual health, risky behaviours including smoking and alcohol use, body image, sleep, peer support, cyberbullying, social media use and the school environment. Items were drawn from the HBSC Survey, the Welsh SHRN and physical activity items from the Youth Activity Profile. In addition, the survey contains the following scales; Short Warwick-Edinburgh Mental Well-being Scale (SWEMWEBS) to assess mental well-being [25], Short Moods and Feelings Questionnaire (SMFQ) to assess moods and feelings [26], the Generalized Anxiety Disorder Scale (GAD-7) to assess anxiety [27, 28], and the Child Health Utility-9 Dimensions (CHU9D) to assess quality of life [29].

The qualitative part is imprecise and unclear, its process and results are not seen.

The qualitative aspect of the study has been detailed within three sections of the manuscript. Firstly, who will be participating in the interviews (section 2.4.2), a description of the sample size (section 2.6.2), and finally a summary of the interview topic guide content (section 2.7.3). As this is a study protocol paper, the manuscript does not contain any qualitative results.

The conclusion is not proved, but rather stated without the reader understanding the process by which this claim is reached.

As this is a study protocol paper, the authors have not provided a conclusion based on results, but a discussion section (Section 4) which summarises what the study aims to achieve, the intended use of the data and possible future use of data generated from the network.

Round 2

Reviewer 3 Report

I thank the authors for clarifying the sections of the study and modifying the indicated recommendations.

I consider the work finally publishable.